# Flow Cytometric Analysis and Sorting of Murine Enteric Nervous System Cells: An Optimized Protocol

**DOI:** 10.3390/ijms26104824

**Published:** 2025-05-18

**Authors:** Faidra Karkala, Indy de Bosscher, Jonathan D. Windster, Savio Stroebel, Lars van Zanten, Maria M. Alves, Andrea Sacchetti

**Affiliations:** 1Department of Pediatric Surgery, Erasmus University Medical Center, Sophia Children’s Hospital, 3015 GD Rotterdam, The Netherlands; f.karkala@erasmusmc.nl (F.K.); j.windster@erasmusmc.nl (J.D.W.); 2Department of Clinical Genetics, Erasmus University Medical Center, Sophia Children’s Hospital, 3015 GD Rotterdam, The Netherlands; l.c.vanzanten@erasmusmc.nl (L.v.Z.); 3Department of Pathology, Josephine Nefkens Institute, Erasmus University Medical Center, 3015 GD Rotterdam, The Netherlands

**Keywords:** ENS, mouse intestine, enteric neurons, enteric glia, FACS

## Abstract

Isolation of neurons and glia from the enteric nervous system (ENS) enables ex vivo studies, including the analysis of genomic and transcriptomic profiles. While we previously reported a fluorescence-activated cell sorting (FACS)-based isolation protocol for human ENS cells, no equivalent exists for mice. As directly applying the human protocol to mouse tissue resulted in low recovery of live ENS cells, we optimized tissue dissociation using mouse colons. A 30 min Liberase-based digestion showed optimal recovery of viable ENS cells, with CD56 and CD24 emerging as the most reliable markers to select and subdivide these cells. ENS’ identity was further validated by FACS, using neuronal (TUBB3) and glial (SOX10) markers and reverse transcriptase quantitative PCR on sorted fractions. Overall, the mouse ENS expression profile significantly overlapped with the human one, showing that current dissociation protocols yield a mixed population of enteric neurons and glia. Nonetheless, using the imaging flow cytometer BD S8 FACS Discover and ELAVL4 as a neuronal soma-associated marker, we observed enrichment of neurons in a CD56/CD24^TIP^ population. In conclusion, we present here a protocol for high-purity FACS-based isolation of viable mouse ENS cells, suitable for downstream applications.

## 1. Introduction

The enteric nervous system (ENS) is a finely tuned network of neurons and glial cells, organized in enteric ganglia within the gastrointestinal (GI) tract. The ENS is located in the myenteric and submucosal plexuses; however, enteric glia are also located in the mucosa and within muscles [1]. The myenteric plexus is found between the circular and longitudinal muscle layers, and the submucosal plexus is situated between the muscle and mucosa [2,3]. The myenteric plexus predominantly regulates muscle contraction, while the submucosal plexus governs fluid secretion and absorption, blood flow, and senses stimuli from the epithelium and lumen, to maintain bowel function [4]. The critical role of the ENS is underscored by the array of enteric neuropathies arising from congenital abnormalities in its development, such as Hirschsprung disease (HSCR) and chronic intestinal pseudo-obstruction [5]. Precise mapping of the ENS and its embryonic development becomes necessary to understand these congenital conditions. However, isolating the ENS poses significant challenges, due to its deep embedding in the GI tract, making it difficult to extract without damaging the delicate ganglionic network [6]. Furthermore, enteric neurons comprise a small proportion of intestinal cells (about 0.01% of the total human colon [6]), and thus ENS enrichment becomes technically demanding.

Recently, we developed a reliable method for isolating human ENS cells, which combined an efficient dissociation protocol with an optimized fluorescent-activated cell sorting (FACS) staining panel, yielding high-quality ENS cells suitable for downstream analysis [7]. Although using human tissue provides the most directly applicable insights into embryonic development and disease mechanisms, it is not always feasible to obtain it. Often, access to human biopsies is limited, especially from embryonic stages, and therefore, using alternative in vivo models, such as mice, allows for a more accessible and consistent source of material. Several mouse models have also been established for studying the ENS as well as enteric neuropathies [8,9,10], contributing to our understanding of genetic complexity and developmental origins of ENS malformations. Previous studies [11] have tried to map the cellular composition of the mouse ENS, using transgenic mice in which ENS cells are labelled. Although this approach allowed for the characterization of numerous neuronal subsets, recapitulating ENS heterogeneity, creating transgenic mice is technically challenging, time-consuming, and expensive. To overcome these issues, a previous study isolated and cultured the myenteric and submucosal plexuses from mice to obtain co-cultures of ENS cells with other non-neuronal intestinal cell types [12]. However, this method was limited because of the altered conditions in which cells grow in vitro leading to biased isolation and characterization of specific neuronal and glial cell subtypes. 

In the current study, we developed a highly specific FACS-based method to efficiently isolate murine ENS cells. We found that CD56 (neural cell adhesion molecule 1 (NCAM1)) and CD24 (HAS; an epithelial protein, but also a neuronal differentiation marker) are able to identify the murine ENS cluster and enrich for neurons vs. glial cells, similarly to what we have described for humans [7]. However, changing the dissociation protocol allowed for the isolation of highly pure and viable enteric neurons and glial cells from murine colonic material, suitable for downstream applications.

## 2. Results

### 2.1. Mouse ENS Cells Are Successfully Isolated with the Human Protocol, but with Low Viability

Based on our previously developed FACS protocol for the isolation and subclustering of human ENS cells [7], we started by exploring its suitability for mouse tissue. Following dissociation of mouse colons with Collagenase II/Dispase, the single-cell suspension obtained was stained with the same cell-surface markers previously used for human cells, only using mouse-reactive antibodies (Table 1) [7]. CD45 (immune-hematopoietic cell marker) and CD31 (immune-hematopoietic and endothelial cell marker) were added as exclusion markers. Cells positive for these markers were referred to as Lin^+^. Red blood cells (RBC) were removed by RBC lysis and DAPI was used to discriminate live from dead cells on the same FACS Aria channel used for the Lin markers (i.e., BV421/DAPI channel, Table 1). Based on our previous results [7], CD56, CD24, and CD90 (Thy-1) were used as positive markers to identify and eventually subdivide the ENS cluster.

In human colon preparations, we have shown that the ENS cluster is composed of both cells and non-nucleated debris, largely overlapping in scattering levels and staining pattern [7]. Moreover, glial cells always carry neuronal remnants, and vice-versa, inevitably leading to a mixed-neuronal/glial staining pattern. With these premises, mouse and human colon samples were analyzed in parallel, using the same gating strategy applied for human tissue [7]. Briefly, following selection of the region of interest (FSC-A vs. SSC-A) with preliminary exclusion of multicellular clusters, as well as very small particles (Figure 1A), a large single-cell gate was applied (FSC-A vs. FSC-W) and live/Lin^−^ events were selected (Figure 1A,B). CD56 expression was then examined to identify the ENS cluster. This marker was selected based on the evidence that it has the highest specificity for ENS cells in human tissue [7]. In mice, a putative ENS cluster was detected with high CD56 expression (CD56^high^) (Figure 1A), which matched well with the human ENS cluster (Figure 1B). A lower CD56 expression (CD56^low^) cluster, more pronounced in mouse samples than in human samples, was also identified, (Figure 1A,B). Next, we examined the co-expression of CD56 with CD24 and CD90 (Figure 1C,D). The CD56^high^ cluster was positive for both CD90 and CD24 in mice, as well as in humans. This evidence supported the hypothesis that CD56^high^ corresponds to the ENS cluster in both systems. In contrast, the CD56^low^ cluster was mostly CD24-negative, suggesting that it is probably not composed of ENS cells. Using CD56, CD24, and CD90 in mice showed that CD56^high^ cells were subclustered in CD24 right (CD24R) and CD24 left (CD24L), and CD90 left (CD90L) and CD90 right (CD90R), respectively (Figure 1E). Human ENS cells showed the expected CD24^low^, CD24^high^, and CD24^TIP^, as well as CD90L and CD90R subclusters (Figure 1F). Since CD24 levels were higher in mouse ENS than in human ENS, we preferred to use the definition of CD24R and CD24L subclusters to differentiate them from the CD24^low^ and CD24^high^ clusters defined in the human system (Figure 1F). Interestingly, our comparison of the CD24L and CD24R to, respectively, the CD90L and CD90R subclusters, did not show correspondence as found in the human ENS (Figure 1E,F). Interestingly, a CD24^TIP^ was also not immediately visible in the mouse ENS, whereas human samples showed, as expected [7], a small but clearly separate CD24^TIP^ subcluster, which roughly corresponded to the upper part of CD90R (Figure 1E,F).

Finally, we investigated the presence, within the ENS cluster, of real ENS cells vs. debris in the mouse samples. Nuclear staining performed with DAPI on sorted and formalin-fixed live/Lin^−^ fractions (Figure 1H) showed that about 35% of the total ENS events belonging to the live fraction were nucleated. Interestingly, this value overlapped with the percentage found in human colons (31 ± 8%). However, further comparison of the percentage of live/Lin^−^ events in human and mouse samples showed that they were significantly lower in mice (Figure 1G). The mean percentage of ENS events, relative to the total number of live/Lin^−^ cells, was also lower when compared to the human system, although this difference alone was not statistically significant. However, when comparing the number of ENS events in the live gate (i.e., the sortable fraction) to the total raw number of events, the former were significantly lower in mouse samples than in human samples (Figure 1G). These results indicated the need to optimize the dissociation protocol for better recovery of the murine ENS.

### 2.2. New Dissociation Protocols for Isolation of Murine ENS Cells

#### 2.2.1. New Dissociation Protocols and Gating Strategy

To improve the dissociation protocol and increase the percentage of viable neural cells isolated, we adjusted its composition and duration. Replacing Collagenase II with Collagenase I was the first adjustment made, as different types of Collagenase break down different types of collagen. By replacing the type of Collagenase and decreasing its concentration, while increasing the concentration of Dispase, we aimed to improve the dissociation efficiency, as described previously [13]. In parallel, we tested a Liberase-based protocol [14], that being a mixture of Collagenase I and II, where Thermolysin replaces Dispase. We also considered that ENS cells are fragile relative to other intestinal cells types, and therefore decreased the dissociation time from 60 to 30 minutes (min). This change was associated with the fact that at 60 min, the samples seemed to be over-digested. In addition, the RBC lysis step was removed to further shorten the protocol and reduce unnecessary stress. As an alternative, an anti-TER119 antibody was added to the Lin antibodies to stain for and gate out all red blood cells.

Prior to any comparison though, we decided to establish a new gating approach. This functioned as quality control for us, to better understand the reasons behind the different outputs potentially observed with different protocols, and to examine the suitability of the alternative digestion approaches (Figure 2 and Appendix A). Since this alternative gating approach was initially established on samples dissociated with Collagenase/Dispase I, the full plots relative to this strategy (including all of the preliminary gates on scattering, DAPI, Lin, and CD56) were obtained using a sample dissociated with Collagenase I/Dispase for 30 min. As a preliminary step, the threshold on FACS was lowered until the smallest DAPI^+^ cells were visible, to make sure all of the dead cells were included in the analysis (Appendix A). Then, we first gated the CD56^high^ particles out of the total (recorded) events, thus including all putative ENS particles that were present in suspension (Figure 2A). These raw events included live and dead ENS cells, debris, and multicellular aggregates. With this approach, we aimed to determine the total fraction of CD56^high^ particles extracted with each dissociation protocol, and identify to which region they corresponded (live, dead, or debris) (Figure 2B). Eventual biases in the ENS distribution introduced by each dissociation protocol could be identified with this approach, e.g., we were able to visualize whether or not missing ENS events in the live region were counterbalanced by an increase in dead cells or small debris. In other words, we could follow the viability and integrity of our putative ENS cells.

In parallel to this unbiased analysis, we followed a modified gating approach to refine our populations (Figure 2C). An initial gate was applied on FSC-A vs. SSC-A to exclude the biggest clusters but include all detectable debris. This was followed by an inclusive “single cell gating” on FSC-H vs. FSC-W, as previously reported [7] and also applied in Figure 1. Notably, the “single cell gate” was calibrated not on all cells, but on the final ENS population, thus including singlets, but also multicellular aggregates until, approximately, the scattering level of triplets/quadruplets. This was necessary to prevent the eventual loss of large and irregular-shaped cells, as dissociated neurons are predicted to be. More stringent doublet exclusion gating, when necessary, was performed at a later stage based on DNA staining [7]. In addition, DAPI^+^ (dead) and Lin^+^ cells were gated separately, as opposed to what was done before, leading to a clear evaluation of Lin^+^ and dead cells. To this purpose, we plotted the DAPI/BV421 channel vs. the free BV510 channel (Figure 2D). Due to a more red-shifted emission spectrum, DAPI showed stronger relative fluorescence in the BV510 than in the BV421 channel when compared to the BV421-conjugated Lin antibodies, making a clear-cut separation of the two fluorochromes possible. Lin^−^ events were further gated in a DAPI/FSC-A plot and subdivided into the following three regions: live, dead, and low-FSC particles or debris. For the sake of clarity, Lin^+^ cells were also shown in parallel to visualize the presence of the expected Lin^+^ clusters (Figure 2E). Overall, this approach allowed us to distinguish low viability from an excess of Lin^+^ cells, and also to draw cleaner live/dead gates. Finally, the live/Lin^−^ events were further analyzed with respect to their CD56, CD24, and CD90 staining pattern to detect the ENS cluster. All of this information, including the qualitative and quantitative elements, was taken into account for a proper comparison of the protocols.

#### 2.2.2. The Liberase-Based Protocol Increases Overall Viability and ENS Recovery from Mouse Colons

Once the strategy was defined, we compared the new Collagenase I/Dispase protocol to a Liberase-based protocol. For a more linear data-flow, we describe, in parallel, FACS plots relative to Collagenase I/Dispase (Figure 3A–D) with Liberase samples (Figure 3E–H), following 30 min digestion. Statistical analysis comparing all of the conditions tested, including 30 min and 1h digestion with Collagenase I/Dispase and Liberase, and the human protocol in parallel, is presented in Figure 3I–K and Appendix A.

Following the preliminary definition of Lin^−^ and live cells, we gated our putative ENS events (Figure 3A,E). The pattern was as expected, with both protocols showing a CD56^high^ and a CD56^low^ cluster. As a second step, we analyzed the expression of CD24 and CD90 (Figure 3B,C,F,G). CD24 was highly expressed by the CD56^high^ cluster and not by the CD56^low^ cluster. The subdivision in CD24R and CD24L was as previously observed, but the CD24R subcluster was less-pronounced in the Liberase samples (Figure 3F vs. Figure 3B). With both protocols, we could detect some ENS particles at very high CD56/CD24 levels, which we defined as CD56/CD24^TIP^ (Figure 3B,F). Although it is not as well-separated as in the human ENS, a density plot confirmed the mouse CD56/CD24^TIP^ to be a small cluster with its own identity. Notably, although the CD24R cluster was smaller in the Liberase samples, the CD56/CD24^TIP^ was comparable to that of the Collagenase samples. Since we expected only the CD24^TIP^ to be enriched with neurons, we did not consider a lower amount of CD24R to be a major issue. CD90 was, as expected, 100% highly expressed by the CD56^high^ cluster (Figure 3C,G). Subdivision in CD90L and CD90R was hardly detectable with the CollagenaseI/Dispase protocol, but it was highly visible with the Liberase protocol. Interestingly, with both protocols there was no overlap of CD24L and CD24R with CD90L and CD90R, respectively (Figure 3D,F), confirming what we observed with the human dissociation protocol on mouse samples (Figure 1E,F). Due to this, and considering that CD90 is not a neuronal selective marker and, also, that common anti-CD90 antibodies do not work with all mouse strains (Table 2), we decided to base our selection and subclustering of the ENS population on mice with only CD56 and CD24.

The full statistical analysis of the different populations and subpopulations clearly showed that the 30 min Liberase protocol significantly increased the percentage of live/Lin^−^ cells, as well as the number of sortable ENS events in the live region (Figure 3I,J). Additionally, the total amount of raw and unbiased ENS particles extracted with the Liberase protocol was significantly higher than with other protocols (Appendix A). Based on these results, we considered the 30 min Liberase protocol to be the most efficient for mouse ENS isolation.

### 2.3. Validation and Subdivision of the ENS Cluster

To validate the identity of the ENS cluster following dissociation with the Liberase protocol, we first performed live nuclear staining to discriminate real ENS cells from ENS debris. This step was essential to define the effective suitability of the dissociation protocol for ENS cell enrichment, as we wanted these cells to be suitable for downstream analysis, such as single-cell RNA sequencing or reverse transcriptase quantitative PCR (RT-qPCR). For this purpose, we added the cell permeant nuclear dye, Dye Cycle Green, to the standard staining panel. Due to this, CD24-BV605 was used instead of CD24-PE, to avoid spillover from the Dye Cycle Green to the PE channel. This set-up allowed us to confirm that the ENS cluster obtained using the Liberase protocol was made of nucleated cells and non-nucleated particles of different sizes (Figure 4A and Appendix A), as previously found with the human protocol. Notably, the average fraction of nucleated ENS cells (34 ± 11%) perfectly overlapped with the one obtained using the human protocol (Figure 1H). This indicates that the fraction of nucleated ENS events out of those gated in the live/Lin^−^ fraction is not significantly influenced by the digestion protocol or the percentage of live cells. On the other hand, prolonged handling, or even simple prolonged storage of ENS samples on ice, can significantly and selectively reduce ENS cells’ viability (Appendix A) and, thus, the percentage of nucleated ENS events in the live region. Interestingly, we also observed that the CD56/CD24^TIP^, considered to be enriched for neurons [7], was more pronounced in the nucleated than in the non-nucleated fraction (Figure 4A–D).

To further validate the identity of the CD56^high^ population, selective neuronal and glial markers were used. In parallel, the CD56^low^ population was also analyzed. We started by using CD24 as a cell-surface/neuron-selective marker within the ENS cluster. Previous analysis (Figure 3E–H and Figure 4A) clearly showed that CD24 expression is associated with the ENS cluster. To better address this point, we conducted an in-depth statistical analysis of those samples (Figure 4D). All ENS particles (cells and debris) were confirmed to be 100% CD24^+^, vs. 8% of CD56^low^ and 22% of CD56^−^ (Figure 4D). The presence of CD24^+^ cells in the CD56^−^ fraction was expected, as epithelial crypt bottom cells are CD24^+^. We then performed a more stringent validation of the ENS population using intracellular staining on formalin-fixed cells, as previously reported [7]. Tubulin Beta 3 (TUBB3/Tuj1) was used as a neuronal selective marker, and SRY-Box Transcription factor 10 (SOX10) was used as a glial selective marker (Figure 4E,F) [11,15]. To discriminate between nucleated and non-nucleated events, and single vs. non-single cells, DAPI was added to the samples. Moreover, gating and interpretation of positive vs. negative populations were made based on negative controls (obtained using matched control antibodies) and fluorescence-minus-one (FMO) samples (Figure 4F). As expected, all putative ENS cells (CD56^high^) were positive for both TUBB3 and SOX10 (Figure 4F,G). On the other hand, CD56^low^ and CD56^−^ cells were negative for Sox10, showing the same background noise (around 10%), as expected when gating positive vs. negative events in the context of a low staining index. Regarding TUBB3 expression, CD56^−^ cells were totally negative, while CD56^low^ showed a small fraction of cells with dim TUBB3 staining. Based on these results, CD56^high^ were confirmed to be ENS cells, whereas the staining pattern of CD56^low^ was not compatible with either glial or neuronal cells. As described for the human samples [7], we observed the presence of a mixed-neuronal and glial staining pattern in all ENS cells, with all of the markers tested. This confirmed that neuronal and glial cells cannot be extracted as pure by any of the dissociation protocols tested by us until now, as they seem to always carry, respectively, glial or neuronal fragments that are attached to them. For the sake of completeness, we have also analyzed the expression of the ENS markers among non-nucleated events. Interestingly, ENS debris already shown to be CD24^+^ (Figure 4G) were also totally positive for TUBB3, indicating that all of them were pure-neuronal or mixed-neuronal/glial fragments, mostly resulting from broken terminations. On the other hand, SOX10 positivity was not universal, indicating the presence of debris that only contains neuronal fragments, and no glial at all (or not enough to stain positive).

Additional validation of the ENS cluster was performed by RT-qPCR using neuronal and glial markers (*Elavl4*; *Tubb3*; *Sox10*; *Cd56/Ncam*). Our results showed that the CD56^high^ population is indeed as enriched for *Cd56* as expected, but also for neuronal and glial markers (Figure 4H). It is important to note that while *Sox10* mRNA expression showed a clear enrichment matching the protein level (FACs data), the fold-change for *Tubb3* mRNA between CD56⁻ and CD56^low^ cells was relatively modest, contrasting with the much stronger difference observed by FACS at the protein level (Figure 4G). This apparent discrepancy can be explained by the cellular composition of the ENS cluster, which is as follows: glial cells represent the vast majority of the CD56^high^/CD24⁺ population, whereas neurons are confined to a very small subset (not higher than 7% according to the TIP subpopulation shown in Figure 4C, but reasonably around 2–3%, as previously reported for the human ENS cluster [7]) located at the CD56^high^/CD24^TIP^ region. Thus, the RT-qPCR result reflects mainly glial transcript levels, where neuronal markers like *Tubb3* are expected to be low and can be further masked by background expression. In contrast, TUBB3 is highly abundant and remains detectable even in neuronal debris attached to glial cells, resulting in a strong signal detected by FACS. Therefore, differences in mRNA and protein levels should be interpreted by considering the technical and biological properties of the isolated cells. To assess the functional integrity of the isolated ENS cells, CD56^high^/CD24⁺ events were sorted and cultured in vitro (Appendix A). Considering that with the BD FACS Aria III platform we were unable to culture viable cells, likely due to the high pressure of sorting, we used the new flow cytometer BD S8 FACS Discover instead, with a very low default pressure on the 130 µm nozzle. This machine combines spectral flow cytometry with cell-view technology, providing spatial and morphological insights on the events under analysis. The gating strategy applied on FACS Discover was similar to the one used on FACS Aria, and it is described in Appendix A. The amount of sorted ENS cells (live and nucleated) was in the range of 40,000 per colon. The morphology of cultured cells was examined under transmitted light, revealing features that are consistent with neuronal and glial cells (Appendix A). To further assess cell identity post-expansion, cultured cells were stained and analyzed using the FACS Aria III. The sustained expression of ENS-specific markers was confirmed, showing that these cells were CD56^/^CD24^+^ (Appendix A). Staining with TUBB3 and SOX10 antibodies further confirmed the ENS identity of these cells (Appendix A).

### 2.4. CD56/CD24^TIP^ Is Enriched in Neurons

In an attempt to subdivide neurons from glial cells, we tested the hypothesis that murine neurons, as human neurons, should have higher levels of CD24 and be enriched in the CD56/CD24^TIP^. This hypothesis was again tested using FACS Discover, as a first approach, by staining for ELAVL4, a selective intracellular neuronal marker concentrated in the neuronal soma and rapidly decreasing to the terminations [7]. Since imaging with FACS Discover is only possible on three blue-laser-dependent channels (green, red, and far red), DAPI was not suitable for imaging, but was still used as a reference nuclear dye. Furthermore, being that the green channel is dedicated to ELAVL4, we decided to visualize the nuclei using the red-emitting Doxorubicin. The latter binds DNA and its excitation and emission spectra, with peaks at 480 and 590 nm, respectively, which matches the second imaging channel well, with little spillover to the first (green) channel. Doxorubicin was further calibrated to minimize its green spillover (Table 2). In addition, to minimize spectral interference of CD24 with the red imaging channel, a near-infrared version of the anti-CD24 antibody (BV786 conjugate) was used.

The gating strategy for formalin-fixed samples on FACS Discover is presented in Figure 5A–D. Briefly, following a preliminary gate on FSC vs. SSC, and a standard, preliminary doublet exclusion strategy, CD56^high^ and CD56^−^ events were selected (Figure 5A). These events were then divided into nucleated and non-nucleated, using DAPI staining (Figure 5A). The nucleated fraction was further subdivided to have single G1/G0 cells, i.e., the region of interest for post-mitotic ENS cells, and a fraction containing a few proliferating and non-single cells (Figure 5B). ENS cells were further subdivided into the TIP vs. non-TIP region (Figure 5C). TIP cells showed high ELAVL4 staining (Figure 5D), bigger shape, and overall higher SSC levels than non-TIP cells, as expected with potential neurons. Statistical analysis of the ELAVL4 median fluorescent intensity (MFI) performed vs. a control antibody showed significantly higher ELAVL4 levels in the TIP (Figure 5E), followed by the rest of the ENS cells. Non-ENS cells had, as expected, the lowest MFI level.Although statistical analysis was only performed on single G1/G0 cells, to correct for different cell sizes we used a ratio approach vs. control Ab, instead of the most common subtraction of control for Ab noise levels.

Imaging of single G1/G0 cells clearly showed the prevalence of ELAVL4 bright cellular shapes compatible with neurons in the TIP, and ELAVL4 dim cells in the ENS/nonTIP region and non-ENS cells (Figure 5F). The region near the TIP (border between TIP and non-TIP) was characterized by intermediate patterns, among them some that resembled ELAVL4^−^ glial cells carrying neuronal fragments (ELAVL4^+^) on their putative proximal part. Imaging with Doxorubicin confirmed that all single cells selected by DAPI had a single red nucleus. On the other hand, the non-nucleated fraction gated with DAPI was confirmed to be devoid of Doxorubicin-positive nuclear-like structures. Interestingly, the CD56/CD24^TIP^ of the non-nucleated fraction was enriched with structures resembling broken neurons containing part of the soma and/or the proximal part of their terminations. Moving to the doublet/triplet regions, it was possible to observe different combinations, among them some ELAVL4^−^ glial cells attached to ELAVL4^+^ neurons. These results confirmed the selectivity of the staining for putative neurons.

## 3. Discussion

In this study, our aim was to develop an optimized protocol for the enrichment and sorting of mouse ENS cells. Our initial approach was to apply the human protocol that we previously developed, based on the use of CD56, CD90, and CD24 markers [7]. Although these markers appeared functional in the identification of putative mouse ENS cells, which were qualitatively similar to the human ones, there were key differences observed. Specifically, the staining pattern observed with CD24 and CD90 appeared to be different in mouse vs. human ENS, highlighting interspecies differences that require tailored experimental approaches. In addition, the viability of the ENS cluster was insufficient for downstream applications. A series of adjustments were, therefore, made to the dissociation protocol, including testing different enzyme combinations and incubation times. Alternating Collagenase II with Collagenase I improved the output, however, it was not significantly. On the other hand, Liberase increased the number of ENS events significantly, but only when the dissociation was performed for 30 min. Extending this process to 60 min led to over-digestion of the tissue, resulting in no ENS yield. Based on these results, we were able to show the importance of fine-tuning protocols to ensure the best outcomes for cell analysis and sorting, particularly in compartments as complex as the ENS.

Further characterization of the ENS cluster obtained was performed to validate the neural nature of these cells. With this purpose, selective markers for neurons (*Cd24*, *Tubb3,* and *Elavl4*) and glial cells (*Sox10*) were used. Interestingly, while RT-qPCRs on sorted samples further confirmed the ENS identity of these cells, the staining pattern obtained was characterized by co-expression of glial and neuronal markers on the same cells, mirroring our findings in the human ENS [7]. This observation points to the intricate relationship between enteric neurons and glial cells, which makes it difficult to extract them as pure populations, at least with the digestion protocols tested until now, since they inevitably carry membrane remnants of each other. As a consequence, defining whether or not a single cell is a neuron or a glial cell carrying proximal or distal neuronal terminations is quite challenging. Nonetheless, by using advanced analysis techniques, such as imaging flow cytometry with the S8 FACS discover, we were able to show that our gating strategy with the selection of CD56/CD24^TIP^ can indeed enrich for neurons vs. glial cells. Taken together, we have developed a FACS-based protocol for the isolation of viable mouse ENS cells, which also allows for preliminary discrimination of neurons from glia.

## 4. Materials and Methods

### 4.1. Animals and Intestinal Isolation

For this study, intestines were removed from C57/BL6 or FVB mice and separated into the small intestine and colon. The latter was placed in a petri dish with cold 1× PBS. Fat was removed and the colon was cut open with a blunt end scissor. Feces was removed from the tissue, which was then placed on a pre-cooled chopping board and gently scraped with a glass slide to remove debris on the luminal side. Finally, the colon was cut into 1 mm pieces with a razorblade.

### 4.2. Dissociation of Mouse Colon Tissue

The pieces were transferred to a gentleMACS C-tube (Miltenyi Biotec, Bergisch Gladbach, Germany, 130-093-237) with 5 mL/mouse of the selected dissociation medium (Table 1), and placed in a gentleMACS Octo Dissociator (Miltenyi Biotec, Bergisch Gladbach, Germany, 130-095-937). The tissue was dissociated for 30 min or 60 min at 37 °C. After dissociation, the suspension was mixed with 5 mL Hanks’ balanced salt solution (HBSS) (Thermo Fisher Scientific, Waltham, MA, USA, 14175095) containing 10% fetal bovine serum (FBS) (Capricorn Scientific, Ebsdorfergrund, Germany, FBS-12A), and gently triturated with a 19G needle and syringe. If needle occlusion was detected, the occluded piece of tissue was removed with a wet wipe and discarded. Then, the suspension was filtered through a 70 μm cell strainer (Falcon^®^, Corning, NY, USA, 352350) and mixed with 10 mL HBSS-10% FBS. Subsequently, the suspension was centrifuged at 4 °C, 400 RCF for 9 min, and resuspended in 2 mL HBSS-10% FBS.

### 4.3. Extracellular and Intracellular Staining

Primary and secondary antibodies, and the corresponding dilutions or final concentrations, are listed in Table 2. Cell suspension was first stained for the selected extracellular markers. The antibody mix was diluted in HBSS 8% FBS, and incubated on ice for 30 min. Cells were then washed twice using HBSS 2% FBS, followed by centrifugation at 4 °C, 540 RCF for 3 min, and resuspension of the pellet in HBSS 8% FBS containing DAPI (1 μg/mL; Sigma-Aldrich, Burlington, MA, USA, D9542) for dead cell exclusion during FACS analysis. For intracellular staining, live cells were incubated with a LIVE/DEAD violet fixable viability dye (Miltenyi Biotech, Bergisch Gladbach, Germany, 130-109-816) for 10 min at room temperature (RT). Then, cells were fixed in 4% paraformaldehyde for 30 min at RT, washed twice with PBS, permeabilized, and blocked with PBS-0.2% Triton X-100, 10% FBS 4 h to O/N. For intracellular staining, both primary and secondary antibodies were diluted in PBS, 0.2% Triton X-100, 10% FBS. For conjugated antibodies, cells were stained on ice for 3 h, washed twice with 1 mL PBS-2% FBS, centrifuged at 4 °C, 540 RCF for 3 min, and resuspended in 100 μL PBS. For unconjugated primary antibodies, cells were washed and stained for an extra 2 h with secondary antibodies, then washed again 3 times.

When staining with secondary Fab monovalent antibodies, 0.25 µg of the primary mouse or rabbit antibody were mixed with 0.75 µg of the secondary Fab at a ratio 1:3, and incubated for 5 min at RT. Pre-dilutions from the antibody stocks were eventually used. The excess of the secondary antibody was then blocked with 4 µL of mouse (Jackson ImmunoResearch, West Grove, PA, USA, 015-000-120) or rabbit (Sigma-Aldrich, Burlington, MA, USA, NS01L-1ML) serum, depending on the secondary antibody used. The mix was immediately diluted in PSS 10% FBS and used for staining, keeping the final concentration of the primary antibody as described in Table 2. Mouse and rabbit control IgGs were pre-incubated with Fab secondary antibodies, and then incubated with cells at the same concentrations used for the corresponding primary antibodies.

### 4.4. Fluorescence-Activated Cell Sorting

FACS analysis and sorting were performed with a BD FACS Aria III or BD S8 FACS Discover (BD Biosciences, Franklin Lakes, NJ, USA), using a 100 micron nozzle for standard analysis and sorting. BD S8 FACS Discover was also used for sorting live cells for in vitro expansion with a 130 µm nozzle and 7 psi pressure. All of the details about the FACS Aria settings and reagents used are reported in Table 2. With FACS Discover, the standard available configuration was used, with 5 lasers and 78 detectors for fluorescence detection. Forward and side scattering were available on both blue and violet laser. Light loss was also available to detect the loss of laser light produced by intercepted particles. Imaging was available on blue-laser only, with 3 channels (green, BP534/46, red, BP 598/60, and far red, BP 788/225) and additional parameters, among them being forward and side scattering, and light loss. Imaging does not allow for compensations between fluorescent channels. Thus, for the sake of simplicity, we limited our FACS Discover usage to simple imaging and small antibody panels to minimize spillover between channels. Furthermore, complex imaging parameters were not used to gate cells, keeping our gating structure close to the one used on FACS Aria III. Scale bar of the images was calculated using 6μm Accudrop beads (BD Biosciences, Franklin Lakes, NJ, USA, 345249) as a reference. Prior to analysis or sorting, the cell suspension was filtered through a 40 μm cell strainer (Falcon^®^, Corning, NY, USA, 352340). Initially, samples were gated by size and granularity using a Side Scatter (SSC-A) versus a Forward Scatter (FSC-A) plot, to allow exclusion of the smallest debris and larger cell clusters. Non-single cells were partially excluded using FSC-H vs. FSC-W gating. The latter gate was not applied stringently, to ensure inclusion of neurons with high FSC. Dead cells were excluded using DAPI, plotted against FSC-A. CD31 (endothelial cells), CD45 (immune cells), and TER119 (red blood cells)-positive cells were also gated out in the same channel as the dead cells to keep other channels free for additional markers. All unstained and compensation controls were applied as previously reported [7].

### 4.5. Gene Expression Analysis

Total RNA was extracted from sorted ENS cells using the RNeasy Micro Kit (QIAGEN, Hilden, Germany, 74034), following the manufacturer’s protocol. Concentration of isolated RNA was determined using the DeNovix DS-11 (DeNovix, Wilmington, DE, USA), and complementary DNA (cDNA) was prepared using the iScript™ cDNA Synthesis Kit (Bio-Rad, Hercules, CA, USA, 1708890 and 1708891), following the manufacturer’s instructions. For the quantification of gene expression, RT-qPCR was performed using iTaq SYBR Green Supermix (Bio-Rad, Hercules, CA, USA, 1725120). For the normalization of gene expression levels, *Gapdh* and *β-Actin* were used as housekeeping genes, and the 2^−∆∆Ct^ method was used for calculations. Primer sets used are listed in Table 3. Three independent experiments were performed to ensure robustness and reliability of the results.

### 4.6. Cell Culture

A 48-well plate was coated with 1:100 Matrigel (Corning^®^, Corning, NY, USA, 356231) and incubated for 1 h at RT. Following this, the well was washed with PBS, and sorted ENS cells (corresponding to ~30,000 events) were plated on matrigel in 1× DMEM/F12 medium containing 5% fetal bovine serum (Serana, Bundoora, VIC, Australia, S-SFBS-EU-015), 100 units/mL Penicillin/Streptomycin (Gibco-Life Technologies, Carlsbad, CA, USA, 15140-122), 20 ng/mL b-FGF (Bio-Techne, Minneapolis, MN, USA, 234-FSE-025), 0.1 mg/mL Normocin (InvivoGen, San Diego, CA, USA, ant-nr-1), 0.002 mg/mL Heparin (Sigma-Aldrich, Burlington, MA, USA, H3149), 6 mM GlutaMAX (Gibco, Waltham, MA, USA, 350500-38), 1× N2 (Gibco, Waltham, MA, USA, 17502048), and 1× B27 (Gibco, Waltham, MA, USA, 17504044). Cells were maintained at 37 °C and 5% CO_2_. Cell morphology was visualized with the EVOS M5000 Microscope (Thermo Fisher Scientific,Waltham, MA, USA).

### 4.7. Quantification and Statistical Analysis

Data were analyzed using GraphPad Prism version 9 (Graphpad Software^®^, La Jolla, CA, USA). Data were tested for normality using the Shapiro–Wilk test. The Mann–Whitney U-test was used if the data were not normally distributed. Differences were tested for statistical significance using either unpaired Student’s *t*-tests or one-way ANOVA. Data are presented as means with standard deviations. Significance thresholds were set as follows: *p* < 0.05 (*), *p* < 0.01 (**), *p* < 0.001 (***), *p* < 0.0001 (****).

## Figures and Tables

**Figure 1 ijms-26-04824-f001:**
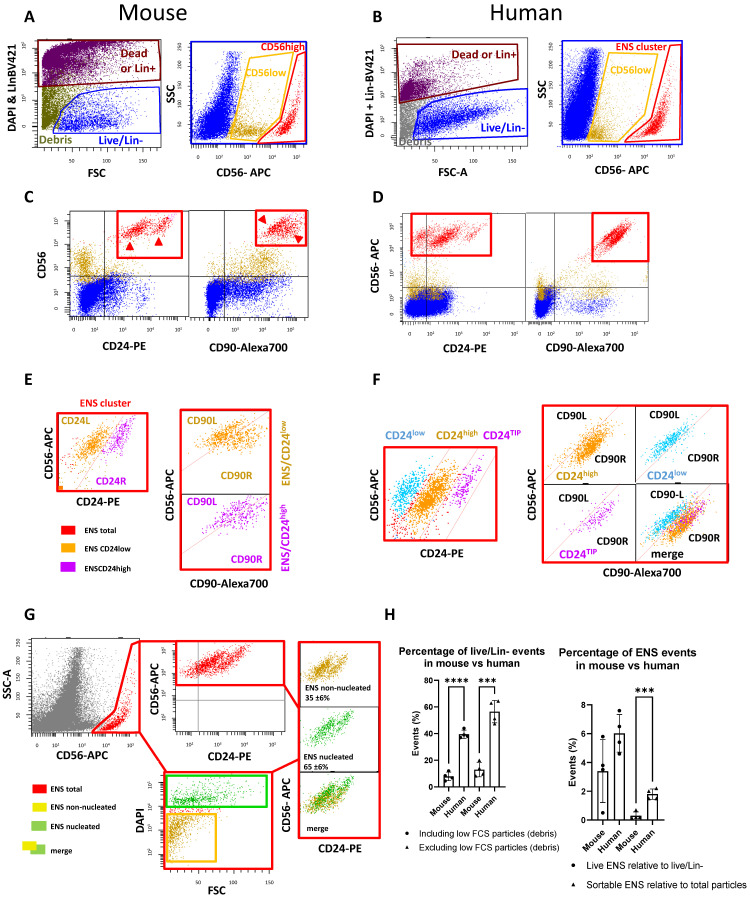
Mouse colon digested using the human dissociation protocol and FACS gating strategy [7] is compared to human colon biopsies. (**A**,**B**) Discrimination of live/Lin^−^ cells (left) and gating on CD56 levels (right) in mouse and human colon, respectively. CD56^high^ (human ENS cluster and putative ENS cluster in mouse) is highlighted in red, CD56^low^ in dark yellow. (**C**,**D**) Expression of CD24 (left plot) and CD90 (right plot) in mouse (**C**) is compared to human (**D**). CD56^high^ and CD56^low^ are in red and dark yellow, respectively. (**E**) Magnification in mouse colon of the CD56^high^ cluster in CD56 vs. CD24 and CD56 vs. CD90 plots (left and right, respectively) shows a division in two subclusters with both CD24 and CD90. In contrast to the human system, the CD24L (left) and CD24R (right) subclusters do not match the CD90L and CD90R subclusters. (**F**) Magnification in human colon of the CD56^high^ cluster in CD56 vs. CD24 and CD56 vs. CD90 plots (left plot and small right plots, respectively) confirms the expected subdivision in three subclusters with CD24, and two with CD90 [7]. Moreover, CD24L (left) and CD24R (right) subclusters match the CD90L and CD90R subclusters. A CD24^TIP^ was also visible on the upper part of the CD90R subcluster. (**G**) The mouse ENS cluster is confirmed to be made of both nucleated and non-nucleated events, largely overlapping in a scattering and staining pattern. Here, live/Lin^−^-sorted mouse cells were fixed and stained with DAPI. The percentage ± SD (N = 3) of nucleated ENS events within the live/Lin^−^ gate was 35 ± 6%, overlapping with that of human colons (31 ± 8%); (**H**) Comparative statistical analysis (mouse vs. human) of the fraction of events falling in the live/Lin^−^ region (left). For the sake of clarity, the percentage of live/Lin^−^ cells has been calculated based on total events after preliminary FCS/SSC and FSC-H|/FSC-W gating (bars on the left side), and “corrected” excluding the debris region from the ratio (bars on the right side). Right: percentage of ENS events out of live/Lin^−^ events and percentage of live sortable ENS events out of the total events. N = 4; Student’s *t*-test. *p*-value (*p*) < 0.001 (***), *p* < 0.0001 (****).

**Figure 2 ijms-26-04824-f002:**
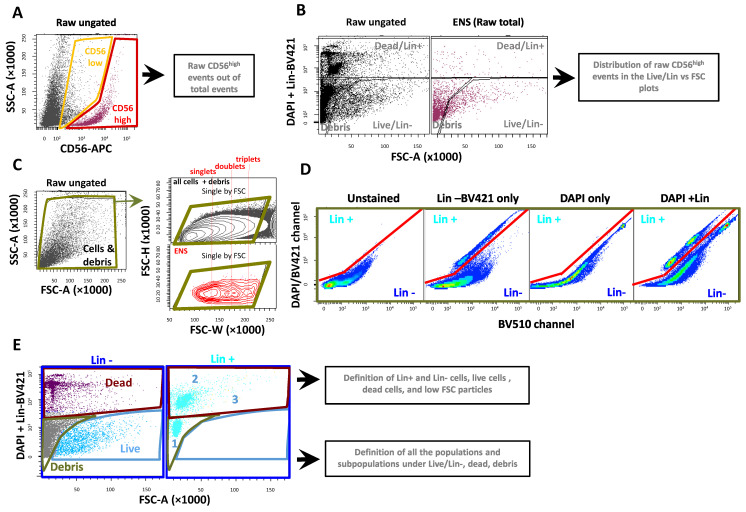
New gating strategy applied to the comparison of different tissue dissociation protocols. (**A**) CD56 plotted vs. SSC-A on the total ungated (raw) events identifies the total (raw) CD56^high^ (ENS) and CD56^low^ particles, independent of whether these belong to live or dead cells, small multicellular clusters, or debris. (**B**) Left plot: based on the distribution of raw ungated events, it was possible to identify three main regions of raw ungated events, as follows: live/Lin^−^, dead or Lin^+^, and low-FSC particles (debris). Right plot: the distribution of raw CD56^high^ events was defined using these three gates. (**C**) Left plot: a preliminary gate was applied to exclude the largest particles (clusters), but keeping in all of the low scattering region. Upper plot: after the first selection, a large single-cell gate was applied on FSC-H vs. FSC-W, as described previously [7]. This gate was calibrated on CD56^high^ events, as shown in the lower plot, to approximately include not only singlets, but doublets and triplets/quadruplets. (**D**) Lin^+^ events were separately gated out by plotting the DAPI/BV421 channel vs. the “free” BV510 channel. Despite the large spectral overlap, plotting the two channels together showed a clear-cut separation of Lin^+^ from DAPI^+^. No compensation was applied here, since the separation of Lin^+^ and dead cells was good enough, while compensation may produce an unwanted deformation of DAPI^+^ events. (**E**) Left plot: selected Lin^−^ cells were further gated in a DAPI/FSC plot and subdivided into live (including nucleated cells), dead, and low FSC particles or debris. Right plot: Lin^+^ events were also plotted in the same frame, leading to the identification of various subclusters. All representative plots were obtained using a sample dissociated with Collagenase I/Dispase.

**Figure 3 ijms-26-04824-f003:**
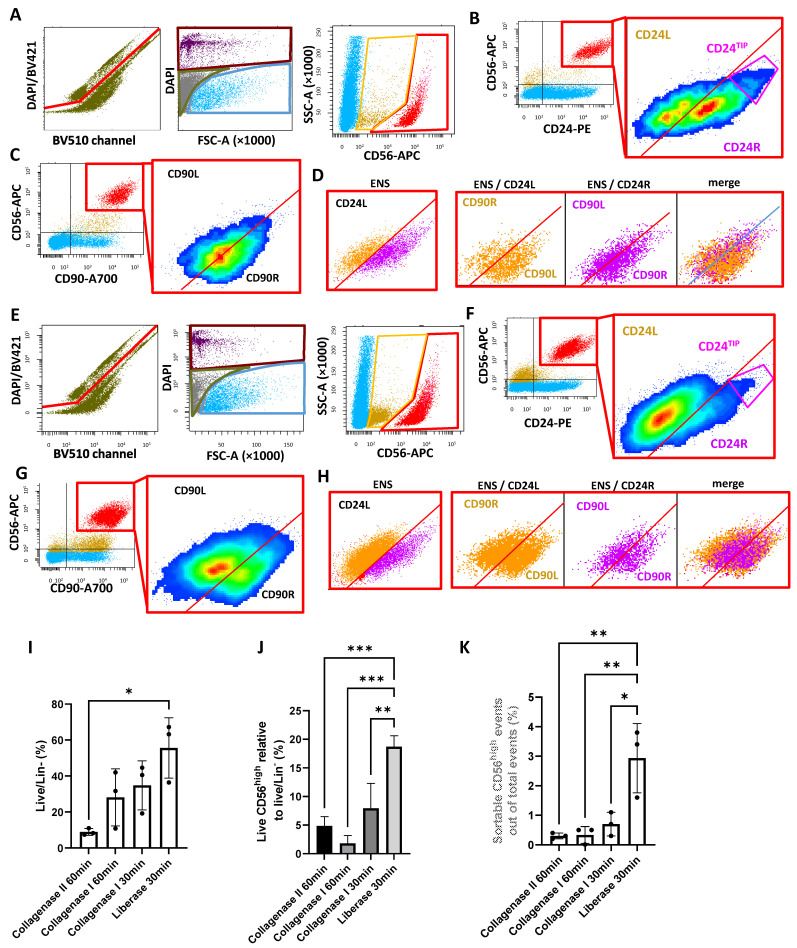
Comparison of alternative tissue dissociation protocols. (**A**) Representative plots showing a sample digested with Collagenase I/Dispase for 30 min. Lin⁺ cells were excluded (left); Lin^−^ cells were gated as live, dead, and debris (middle); and the CD56^+^ cluster was identified (right). (**B**) CD24 vs. CD56 plot (left) reveals a CD24^high^ ENS cluster. Right inset: magnification of the CD56^high^/CD24^high^ cluster using a density plot. This cluster can be subdivided into two main regions (CD24L, CD24R) and has a potential CD56/CD24^TIP^ population. (**C**) Right plot: CD90 vs. CD56 (left) identifies a CD56^high^/CD90^high^ ENS cluster. Right: density plot showing magnification of the cluster and tentative subdivision into CD90L and CD90R regions, though separation was not clearly defined with this protocol. (**D**) As in Figure 1C, the CD24-based ENS subclusters are plotted in CD90 vs. CD56, confirming no match between CD24L/CD24R and CD90L/CD90R. (**E**–**H**) The same analysis shown in (**A**–**D**) is repeated here for Liberase-digested samples, with the following two differences: (1) the CD24R subcluster is smaller than CD24L; (2) CD90 shows clearer separation between L and R, though variability exists across samples. (**I**–**K**) Summary statistics (N = 3; Student’s *t*-test) comparing Liberase and Collagenase I/Dispase protocols, including the human protocol (Collagenase II/Dispase) as a reference. *p* < 0.05 (*), *p* < 0.01 (**), *p* < 0.001 (***).

**Figure 4 ijms-26-04824-f004:**
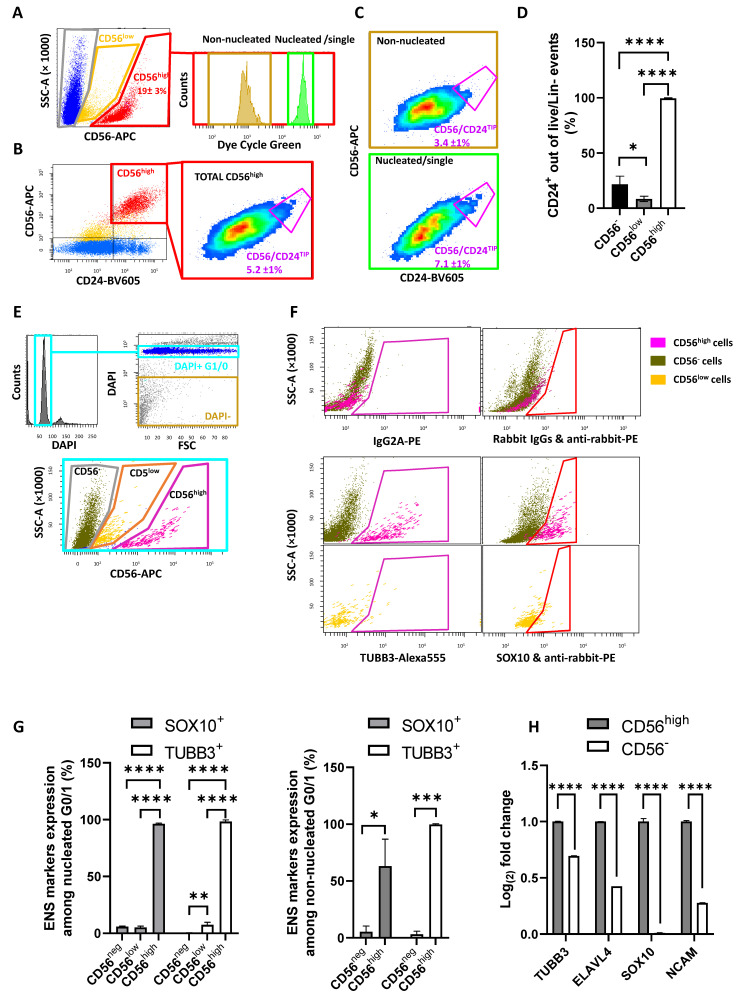
Liberase-dissociated sample is analyzed to define the presence of ENS cells vs. debris, to identify the CD56/CD24^TIP^, and to validate the ENS identity of CD56^high^ cells vs. CD56^−^ and CD56^low^ cells. (**A**) After gating on live/Lin^−^, CD56^high^ and low events were identified (left) and subdivided into nucleated vs. non-nucleated (right) by means of live nuclear staining with Dye Cycle Green. (**B**) In parallel, the whole CD56^high^ cluster is visualized in a CD24 vs. CD56 plot (left) and investigated for the presence of a CD56/CD24^TIP^ region (right, big inset). CD24-BV786 was used, instead of CD24-PE, to minimize the spillover from Dye Cycle Green to the PE channel. (**C**) Crossing information from (**A**,**B**), both non-nucleated debris (upper density plot) and nucleated CD56^high^ cells (lower density plot) showed the presence of a CD56/CD24^TIP^. The latter is more pronounced in the nucleated fraction (percentage ± SD, N = 4, is reported inside the plots). (**D**) The percentage of CD24^+^ cells in the different fractions is reported (N = 4; one-way ANOVA). Being expressed on epithelial cells, CD24^+^ cells are expected to be present in the CD56^−^ subcluster. Notably, CD56^low^ cells had a lower CD24^+^ fraction than CD56^−^. (**E**) Samples pre-stained with CD56-APC were formalin-fixed and post-stained for TUBB3 and SOX10. Following DAPI staining, nucleated single cells in G1/0 were selected (upper plots), and CD56 regions were selected. (**F**) Cells present in the CD56 regions were investigated for the expression of TUBB3/Tuj1 (left) as a neuronal marker, and SOX10 (right) as a glial marker. CD56^high^ and CD56^−^ cells are in the upper plots in magenta and dark green, respectively. CD56^low^ cells are represented alone in yellow (lower plots). (**G**) Histogram plots showing statistical analysis of TUBB3 and SOX10 expression (N = 3). On the left, nucleated cells are compared (one-way ANOVA). On the right, non-nucleated debris fractions are shown (Student’s *t*-test). (**H**) RT-qPCR performed on samples collected from sorted live mouse ENS cells vs. non-ENS fraction showed that neuronal and glial markers are higher expressed in the ENS cluster (N = 3; Student’s *t*-test). *p* < 0.05 (*), *p* < 0.01 (**), *p* < 0.001 (***), *p* < 0.0001 (****).

**Figure 5 ijms-26-04824-f005:**
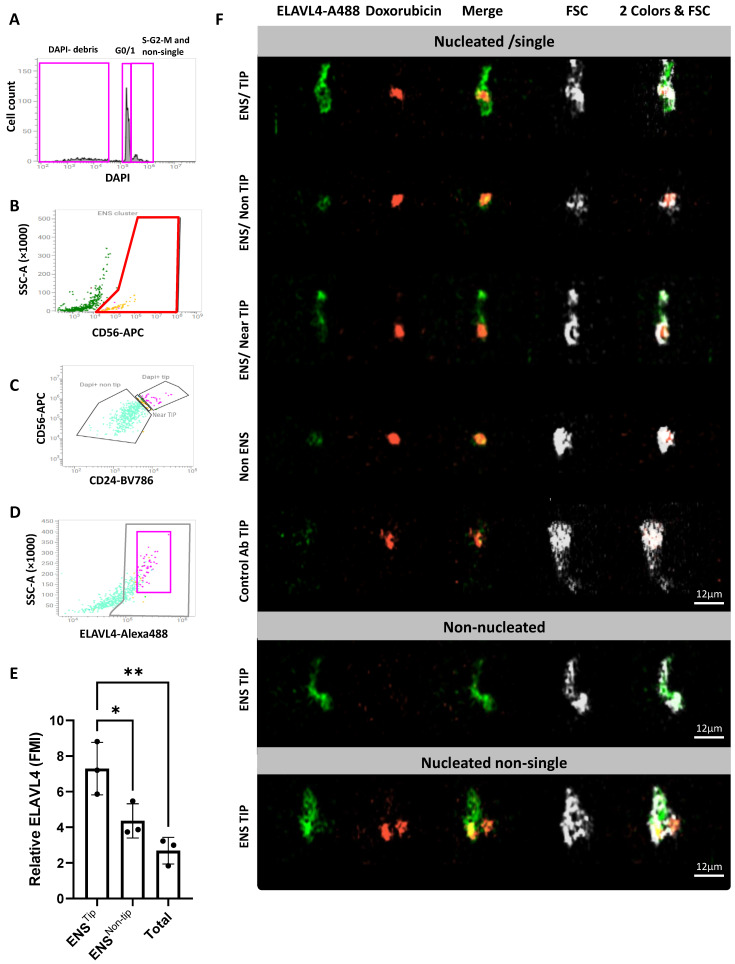
Formalin-fixed ENS cells, pre-stained for CD56 and CD24, were stained for ELAVL4. (**A**) Based on DAPI staining, non-nucleated debris were divided from nucleated cells, as shown in Figure 4E. The latter were subdivided, by crossing DAPI levels with a DAPI-A vs. DAPI-W plot in a cluster of interest containing single cells in G1/0, single cycling cells in S-G2/M, and multicellular aggregates. (**B**) After doublet exclusion, ENS cells were selected. Non-nucleated particles were similarly gated. (**C**) ENS cells were visualized in a CD56 vs. CD24 plot, to gate ENS TIP vs. non-TIP. A non-nucleated ENS TIP was also gated and analyzed separately, as in Figure 4C. (**D**) The ENS TIP appeared at higher ELAVL4 and SSC levels than other ENS cells, i.e., compatible with neurons. The ELAVL4^+^ region, defined based on a control antibody, is gated in gray. (**E**) Statistical analysis of ELAVL4 levels, measured as MFI in ENS TIP vs. ENS non-TIP and non-ENS cells (N = 3; one-way ANOVA). To correct for different cell size, potentially influencing the total ELAVL4 fluorescence, the raw MFI of each population was divided by the corresponding MFI obtained with a control antibody. (**F**) Visualization of the cells under analysis using a BD S8 FACS Discover, equipped with the new cell-view technology. ELAVL4 is in green, and Doxorubicin, used to visualize the nuclei, is in red. FSC-imaging was used here as a microscopy brightfield-equivalent, to visualize the cells and debris under analysis. The ENS TIP is enriched with ELAVL4^bright^ putative neurons. The ENS non-TIP is mostly composed of ELAVL4^dim^ or ELAVL4^−^ cells, associated with ELAVL4^dim^ (neuronal) terminations. Intermediate figures can be observed at the border between ENS TIP and non-TIP (see gap in c), e.g., putative glial cells carrying proximal neuronal terminations, as shown on the third row. Overall, these data point to the existence of a gradient of ELAVL4^bright^ to ELAVL4^dim^ figures, which move down from the ENS TIP to the ENS bottom, most of them being glial cells associated with proximal neuronal fragments, and, finally, distal terminations. *p* < 0.05 (*), *p* < 0.01 (**).

**Table 1 ijms-26-04824-t001:** Dissociation media.

Components	Manufacturer	Cat. No.	Final Concentration	Dissociation Medium ^1^
DMEM/F12	Gibco (Waltham, MA, USA)	11320-074	N/A	A & B & C
HEPES (1M)	Thermo Fisher Scientific (Waltham, MA, USA)	15630106	10 mM	A & B & C
DNase I	Sigma-Aldrich (St. Louis, MO, USA)	11284932001	200 µg/mL	A & B & C
Dispase	Gibco (Waltham, MA, USA)	17105-041	0.25 mg/mL	A
Dispase	Gibco (Waltham, MA, USA)	17105-041	1 mg/mL	B
FBS	Capricorn Scientific (Ebsdorfergrund, Germany)	FBS-12A	5%	B
Collagenase II	Gibco (Waltham, MA, USA)	17101-017	3mg/mL	A
Collagenase I	Gibco (Waltham, MA, USA)	17101-015	1 mg/mL	B
Liberase	Roche (Basel, Switzerland)	5401119001	0.5 mg/mL	C

^1^ A: Collagenase II/Dispase medium; B: Collagenase I/Dispase medium; C: Liberase medium.

**Table 2 ijms-26-04824-t002:** FACS Aria settings and antibodies.

**Fluorophore**	**Lasers**	**BP Filter (nm)**	**LP Filter (nm)**	
Hoechst/DAPI/BV421	405 nm	450/40	-	
FITC/A488/Cycle Green	488 nm	530/30	502	
PE	561 nm	582/15	-	
APC	633 nm	660/20	-	
Alexa 700	633 nm	730/45	690	
BV605	405 nm	610/20	570	
BV786	405 nm	780/60	750	
BV510	405 nm	530/30	502	
**Primary Antibody**	**Reactivity ^1^**	**Fluorochrome**	**Supplier; Cat.#**	**Dilution**
CD56	H	APC	Biolegend (San Diego, CA, USA); 362504	1:40
CD90	H	Alexa 700	Sony Biotechnology (San Jose, CA, USA); 2240600	1:40
CD24	H	PE	BD (Franklin Lakes, NJ, USA); 555428	1:20
CD31	H	BV421	Biolegend (San Diego, CA, USA); 564089	1:0
CD45	H	BV421	Biolegend (San Diego, CA, USA); 304031	1:40
CD24	M	PE	Biolegend (San Diego, CA, USA); 101807	4 µg/mL
CD24	M	BV786	BD (Franklin Lakes, NJ, USA); 744470	4 µg/mL
CD24	M	BV605	Biolegend (San Diego, CA, USA); 101827	4 µg/mL
CD56	M	APC	R&D systems (Minneapolis, MN, USA); FAB7820A-100UG	4 µg/mL
CD90	M	Alexa 700	Biolegend (San Diego, CA, USA); 105320	4 µg/mL
CD45	M	BV421	Biolegend (San Diego, CA, USA); 103133	4 µg/mL
CD31	M	BV421	Biolegend (San Diego, CA, USA); 102423	4 µg/mL
TER119	M	BV421	BD (Franklin Lakes, NJ, USA); 563998	4 µg/mL
TUBB3	M/H	Alexa 555	BD (Franklin Lakes, NJ, USA); 560339	1 µg/mL
TUBB3	M/H	Alexa 488	Biolegend (San Diego, CA, USA); A488-435L	1 µg/mL
SOX10	M/H	Unconjugated	ThermoFisher (Waltham, MA, USA); 10422-1-AP	0.4 µg/mL
ELAVL4	M/H	CL 488	Proteintech (Rosemont, IL, USA); 67835-1-Ig	1 µg/mL
Control Mouse IgG	-	Unconjugated	Santa Cruz Biotechnology (Dallas, TX, USA); sc-2025	1 µg/mL
Control Rabbit IgG	-	Unconjugated	Biolegend (San Diego, CA, USA); 910801	0.4 µg/mL
Control Mouse IgG2a		PE	BD Pharmingen™; 555574	1 µg/mL
**Secondary Antibody**	**Reactivity**		**Supplier; Cat.#**	**Dilution**
Goat anti-rabbit (Fab)	Rabbit IgGs	PE	Jackson ImmunoResearch (West Grove, PA, USA); 111-117-008	-
Donkey anti-mouse (Fab)	Mouse IgGs	Alexa 488	Jackson ImmunoResearch (West Grove, PA, USA); 715-547-003	-
**Nuclear Staining**				
Dye Cycle™ Green	-	-	ThermoFisher (Waltham, MA, USA); V35004	1:4000 ^2^
Doxorubicin	-	-	Pharmachemie (Haarlem, The Netherlands), 51.223.805	0.4 µg/mL ^2^
DAPI	-	-	Sigma-Aldrich (St. Louis, MO, USA)	1 µg/mL ^2^

^1^ M: mouse-reactive; H: human-reactive. Being irrelevant in this context, cross-reactivity with other species is not mentioned. ^2^ The indicated concentration is suitable for samples with ≤1 × 10^6^ cells/mL.

**Table 3 ijms-26-04824-t003:** Primers used for RT-qPCR.

Gene	Forward Primer (5→3)	Reverse Primer (5→3)
*Tubb3*	CAGATAGGGGCCAAGTTCTGG	GTTGTCGGGCCTGAATAGGT
*Elavl4*	TCAGACTCCAGACCAAAACCA	TGATGCGACCGTATTGAGAGA
*Sox10*	GCAAGACACTAGGCAAGCTC	CCTCTCAGCCTCCTCAATGA
*Ncam1*	CACCATCTACAACGCCAACA	GGGGTTGGTGCATTCTTGAA
*β-Actin*	CTCCACCAGTCTTAAATGGA	AACATAACAACTCTGCAGTCA
*Gapdh*	ACTTTGGCATTGTGGAAGGG	ACAGTCTTCTGGGTGGCAGTG

## Data Availability

Further information and requests for reagents may be directed to, and will be fulfilled by, Maria M. Alves (m.alves@erasmusmc.nl) and Andrea Sacchetti (a.sacchetti@erasmusmc.nl).

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
