# Peer review of "Flow Cytometric Analysis and Sorting of Murine Enteric Nervous System Cells: An Optimized Protocol"

_ijms, 2025, doi:10.3390/ijms26104824_

Round 1
Reviewer 1 Report
Comments and Suggestions for Authors
Dear Authors,
Congratulations on very interesting article. I liked the way you described your research. A full description of the process you went through to optimize ENS cell isolation. You are conducting the interesting and very necessary research because it turns out that the intestines, their microbiota, and their nervous system have a significant impact on the health of animals and humans. Connecting the study of ENS cells with the study of the gut microbiota would be a fantastic continuation of this type of research.
When it comes to the article. The introduction is well written and comprehensive. It explains well the need for the research.
Materials and methods.
The procedures have been thoroughly described with all the necessary details. The experiments were carefully planned with all the necessary cytometric procedures and controls. Gating strategies for obtaining ENS cells are clear and precise.
The results are interesting and clearly presented with cytograms, graphs and imaging pictures.
The discussion covers all important aspects of research and describes possible shortcomings of the isolation which is, in my opinion, a great advantage of this study.
The article is well written without major issues to correct. I have only some minor suggestions concerning the article that I noticed reading the manuscript.
When describing reagents and materials the name of the company, city and country is usually put in brackets after the name of the reagent. Catalogue numbers could be included in the separate table. However, personally I would prefer your way of reagents’ description, because scientists are more interested in cat. No. that the city and the country of the producer.
The Authors have used median fluorescence intensity instead of mean fluorescence intensity, the strategy I fully recommend and use myself in flow cytometric investigations. I think it better characterizes cell populations that mean fluorescence intensity. But in line 410 the Authors use abbreviation Fluorescence Median Intensity (FMI), and in other lines median fluorescence intensity (MFI) with lower case. Please, unify that throughout the text. I would rather use MFI abbreviation.
Line479 paragraph Intracellular and extracellular staining is lacking centrifugation parameters – time, rpm
Line 431 specie, which require tailored approaches.
Line 436, 437 However, this was only the case when the dissociation was performed for 30’.
60‘. I would rather use min that apostrophe. It is more clear for the reader.
In the Table 2. FACS Aria settings and antibodies, column Supplier; Cat.#, please unify the description (commas, semicolons, etc.)
Author Response
We thank the reviewer for reading the manuscript and for the insightful comments. Please find below a point-by-point response to your suggestions.
Comment Reviewer 1: When describing reagents and materials the name of the company, city and country is usually put in brackets after the name of the reagent. Catalogue numbers could be included in the separate table. However, personally I would prefer your way of reagents’ description, because scientists are more interested in cat. No. that the city and the country of the producer.
Authors’ response: We appreciate your acknowledgment of our current format. Given the absence of specific formatting requirements from the journal, we have retained our original style for reagent descriptions. However, we remain open to adjustments should the editorial team requests a standardized format.
Comment Reviewer 1: The Authors have used median fluorescence intensity instead of mean fluorescence intensity, the strategy I fully recommend and use myself in flow cytometric investigations. I think it better characterizes cell populations that mean fluorescence intensity. But in line 410 the Authors use abbreviation Fluorescence Median Intensity (FMI), and in other lines median fluorescence intensity (MFI) with lower case. Please, unify that throughout the text. I would rather use MFI abbreviation.
Authors’ response: We appreciate your acknowledgement of our strategy to use the Median Fluorescence Intensity (MFI) for a more accurate representation of our data. In accordance to this comment, we have unified the abbreviation of MFI throughout the text.
Comment Reviewer 1: Line479 paragraph Intracellular and extracellular staining is lacking centrifugation parameters – time, rpm.
Authors’ response: In accordance to this comment, we have included the centrifugation parameters. Please see now lines 538-539.
Comment Reviewer 1: Line 431 specie, which require tailored approaches.
Authors’ response: In accordance to this comment, we have revised the sentence to improve clarity. The updated sentence reads now: “Specifically, CD90 appeared to not be useful in mice, highlighting interspecies differences that require tailored experimental approaches.”
Comment Reviewer 1: Line 436, 437 However, this was only the case when the dissociation was performed for 30’.
Authors’ response: We have revised the sentence to improve clarity. Please see lines 490-491.
Comment Reviewer 1: 60‘. I would rather use min that apostrophe. It is more clear for the reader.
Authors’ response: In accordance to this comment, we have changed the apostrophe with “min”.
Comment Reviewer 1: In the Table 2. FACS Aria settings and antibodies, column Supplier; Cat.#, please unify the description (commas, semicolons, etc.)
Authors’ response: In accordance to this comment, we have unified the format on the description in Table 2.
Reviewer 2 Report
Comments and Suggestions for Authors
The manuscript by F. Karkala et al. presents a novel approach to flow cytometry analysis of enteric nervous system (ENS) cells. The authors have thoroughly optimized tissue dissociation methods using various enzyme combinations and propose a new gating strategy for isolating ENS cells. This study builds upon their previous work, now adapted to the murine model. While the manuscript is well-structured and the findings are compelling, I have a few comments and suggestions for improvement:
- The current title may be somewhat misleading. Although I understand the rationale behind the "from human to mouse" phrasing, it might imply that the manuscript methodology or analyses related to the human ENS, which is not the case. A clearer title would help set the right expectations for readers.
- The resolution of some figures is relatively low, making axis labels and details difficult to read. It’s unclear whether this issue is due to the PDF rendering or the original figure files. The authors may want to check the figure quality and ensure high-resolution images are provided for clarity.
- In section 2.2.1, the discussion on the Liberase-based protocol would be more appropriately placed in section 2.2.2. As it stands, the reader may assume that the Liberase protocol is the preferred or primary method, yet the authors proceed to use the Collagenase/Dispase I protocol for the remainder of the experiments and figures. Relocating lines 162–169 to section 2.2.2 would improve the logical flow and clarity of the methods.
- Line 237-238: The sentence “CD24R subcluster was less pronounced in the Liberase samples (Figures 3f vs 1b)” appears to have a citation error. It seems the correct comparison should be Figures 3f vs 3b.
- In section 2.3 (from line 309): The wording in this section should be revised for clarity. Based on the current data, the authors cannot definitively state that CD56⁻ cells are negative for Sox10 and TubB3, while CD56low cells contain a small TubB3⁺ population. As presented, CD56⁻ for Sox10 appears comparable to CD56low for TubB3. Therefore, the conclusion that one population is Sox10-negative and the other TubB3-positive is not sufficiently supported.
- Figure 4G: Both graphs are labeled with the same axis. I assume, based on the text, one graph represents ENS marker expression among nucleated G0/1 cells, and the other shows ENS marker expression among non-nucleated fragments. If that is correct, the figure should be updated for clarity.
- Figure 4G: The finding that 100% of cells are positive for both Sox10 and TubB3 raises some concerns. Figure 5 appears to support the idea that this population is purely neuronal, and that Sox10 detection may stem from glial membrane fragments adhering to neurons. However, the RT-qPCR data in Figure 4H show that Sox10 mRNA is indeed present, implying that it is not only membrane debris. Did the authors attempt immunofluorescence or another imaging method to confirm whether Sox10 is expressed in intact cells?
- Figure 4: In Figure 4H, the fold-change difference in TubB3 expression between CD56⁻ and CD56low cells is modest—approximately 2-fold—whereas in Figure 4G, the difference in TubB3+ cells appears to be around 20-fold. However, the data for Sox10 mRNA and Sox10+ cells are coherent. Can the authors provide an explanation for this discrepancy between mRNA and protein levels of TubB3?
- Minor suggestion: the bibliography about the ENS is very old and could be updated with more recent papers (citations 1,3, and 8)
Author Response
We thank the reviewer for carefully reading the manuscript and providing insightful comments. These helped us to further improve this work. Please find below a point-by-point response to your suggestions.
Comment Reviewer 2: The current title may be somewhat misleading. Although I understand the rationale behind the "from human to mouse" phrasing, it might imply that the manuscript methodology or analyses related to the human ENS, which is not the case. A clearer title would help set the right expectations for readers.
Authors’ response: We appreciate the insightful feedback regarding the potential misinterpretation of the title. The reference to “human” originates from two key aspects: first the protocol used in this study was adapted from a previously published method developed for human tissue; second, the study includes data derived from human samples, which are presented in Figure 1. To avoid confusion and better reflect the scope of the study, we have revised the title accordingly. The new title reads: “Flow cytometric analysis and sorting of murine enteric nervous system cells: an optimized protocol”.
Comment Reviewer 2: The resolution of some figures is relatively low, making axis labels and details difficult to read. It’s unclear whether this issue is due to the PDF rendering or the original figure files. The authors may want to check the figure quality and ensure high-resolution images are provided for clarity.
Authors’ response: We agree with the reviewer about the necessity of high-resolution images, therefore we have managed to increase the resolution hoping that all details are now visible. However, we remain open to adjustments should the current format still be unsatisfactory.
Comment Reviewer 2: In section 2.2.1, the discussion on the Liberase-based protocol would be more appropriately placed in section 2.2.2. As it stands, the reader may assume that the Liberase protocol is the preferred or primary method, yet the authors proceed to use the Collagenase/Dispase I protocol for the remainder of the experiments and figures. Relocating lines 162–169 to section 2.2.2 would improve the logical flow and clarity of the methods.
Authors’ response: In section 2.2.1, we focus on different dissociation protocols, but also on the new gating strategy. Therefore, we believe that the introduction for the Liberase based method is well placed. However, we have unfortunately not conveyed our message in the appropriate way and have now changed section 2. Please see lines 180-184. Briefly, the new gating strategy, which provides more insights into qualitative and quantitative difference between samples derived by different dissociation protocols, is a preliminary step before comparing Collagenase I vs Liberase, and 30 min vs 1h incubation. Essentially, the plan was to optimize our gating strategy for a multi-comparison, which included different protocols, also liberase. However, the gating startegy was initially tested on Collagenase I/Dispase and Collagenase II/Dispase samples, before it became clear that Liberase was the best protocol. This is the reason why we report, as an example, Collagenase I/Dispase samples in Fig 2 . It should be clear that it is just an example of preliminary gating strategy that was applied later to all the digestion protocols included in the comparison. In Fig 3 and Fig. S1 we apply this strategy to all the digestion protocols, reporting also downstream detailed plots including CD56, CD24 and CD90, which are not present in Figure 2, since the latter is only for the sake of clarity. This point is also now explained in section 2.2.
Comment Reviewer 2: Line 237-238: The sentence “CD24R subcluster was less pronounced in the Liberase samples (Figures 3f vs 1b)” appears to have a citation error. It seems the correct comparison should be Figures 3f vs 3b.
Authors’ response: We have corrected the error, as indeed it should have been 3b instead of 1b.
Comment Reviewer 2: In section 2.3 (from line 309): The wording in this section should be revised for clarity. Based on the current data, the authors cannot definitively state that CD56⁻ cells are negative for Sox10 and TubB3, while CD56low cells contain a small TubB3⁺ population. As presented, CD56⁻ for Sox10 appears comparable to CD56low for TubB3. Therefore, the conclusion that one population is Sox10-negative and the other TubB3-positive is not sufficiently supported.
Authors’ response: We thank the reviewer for this thoughtful observation. As we understand the comment, the reviewer points out that the fluorescence intensity range for CD56⁻ cells stained for Sox10 appears comparable to that of CD56low cells stained for Tubb3, which raises concerns about definitively classifying one population as SOX10-negative and the other as TUBB3-positive. To address this, we have updated Figure 4 to include the relevant negative controls and fluorescence-minus-one (FMO) samples for both stainings, with the corresponding text in lines 324-327. As specified in the Material and Methods, gating was performed using unstained controls and isotype control antibodies, processed under identical staining conditions. For the TUBB3 staining, we used a one-step protocol with a directly conjugated mouse IgG2a Alexa Fluor 555 monoclonal antibody. As a control, we used a mouse IgG2a isotype antibody conjugated to PE. These controls confirmed that the CD56high population showed clear TUBB3 positivity with the specific antibody, while the CD56low and CD56⁻ populations remained at or near background levels. For SOX10, the staining was performed using an unconjugated primary antibody followed by a secondary antibody in a two-step protocol. As expected, this method may result in higher background due to potential variability in permeabilization, blocking efficiency, or incubation conditions. We have now included the isotype control for SOX10 as well, which supports our interpretation that only the CD56high population demonstrates specific Sox10 staining (new Figure 4).
Comment Reviewer 2: Figure 4G: Both graphs are labeled with the same axis. I assume, based on the text, one graph represents ENS marker expression among nucleated G0/1 cells, and the other shows ENS marker expression among non-nucleated fragments. If that is correct, the figure should be updated for clarity.
Authors’ response: We thank the reviewer for this point. We have corrected the legend of the second plot, which refers to non-nucleated particles.
Comment Reviewer 2: Figure 4G: The finding that 100% of cells are positive for both Sox10 and TubB3 raises some concerns. Figure 5 appears to support the idea that this population is purely neuronal, and that Sox10 detection may stem from glial membrane fragments adhering to neurons. However, the RT-qPCR data in Figure 4H show that Sox10 mRNA is indeed present, implying that it is not only membrane debris. Did the authors attempt immunofluorescence or another imaging method to confirm whether Sox10 is expressed in intact cells?
Authors’ response: We thank the reviewer for this insightful comment. We would like to clarify that Figure 5 demonstrates that the CD56/CD24TIP subpopulation is enriched for enteric neurons. However, the broader CD56high/CD24 population — which defines the ENS cluster — is composed primarily of glial cells, with neuronal cells representing only a small fraction of this population. The CD56/CD24-TIP, which we have demonstrated to be enriched with neurons, is around 7% of the nucleated ENS cells (Figure 4a, 4b). However, if neurons are enriched in the TIP, this doesn’t mean the TIP is a purely composed by neurons. We believe this is a large overestimation, as the % of neurons present is realistically around 3% in optimal samples. We have well demonstrated this point in our previous manuscript using human samples [1], in which a well separated CD24-TIP cluster was identified [1]. Moreover, in the same paper we have also shown that the % of viable neurons recovered in a single cell RNAseq assay after sorting, is 1.6% in average, suggesting some loss of neuronal viability due to the harsh sorting procedures and the fragile nature of dissociated neurons in suspension, after being separated from their terminations. Moreover, similar numbers have been shown in the murine system as well [2]. Therefore, the presence of SOX10 mRNA detected by RT-qPCR in Figure 4H is consistent with the predominance of glial cells in this population. On the other hand, the expression of neuronal markers such as TUBB3 and CD24 observed by flow cytometry is likely influenced by the presence of neuronal debris which remain adherent to glial cells after the dissociation process. As a result, we observe relatively high protein-level detection (by FACS), despite the low proportion of true neurons — a phenomenon that has been previously described and that we also observed in human ENS samples [1].
Comment Reviewer 2: Figure 4: In Figure 4H, the fold-change difference in TubB3 expression between CD56⁻ and CD56low cells is modest—approximately 2-fold—whereas in Figure 4G, the difference in TubB3+ cells appears to be around 20-fold. However, the data for Sox10 mRNA and Sox10+ cells are coherent. Can the authors provide an explanation for this discrepancy between mRNA and protein levels of TubB3?
Authors’ response: We thank the reviewer for this valuable observation, although we have to remark that CD56low cells are not reported in Figure 4H, so we will refer to CD56- cells. As noted, the apparent discrepancy between the modest (~2-fold) difference in Tubb3 mRNA expression (Figure 4h) and the more pronounced difference in TUBB3 protein-positive cells observed by flow cytometry (Figure 4g) is indeed notable. This divergence can be explained by the cellular composition of the CD56⁺/CD24⁺ ENS cluster. Glial cells represent the overwhelming majority of this population, with neurons restricted to a small subset (~1–3%, as discussed above) located at the CD56/CD24TIP region. This proportion has been consistently confirmed in our previous single-cell RNA sequencing analyses of human ENS [1], as well as in the murine system [2]. When analyzing the entire ENS cluster by RT-qPCR, we are essentially measuring transcript levels from an almost purely glial population. This explains the robust Sox10 signal and the comparatively low Tubb3 mRNA levels. Additionally, due to the small size of the neuronal subset, any low-level expression of Tubb3 in non-neuronal cells or minimal contamination from attached neuronal fragments can contribute to background noise, potentially masking the true fold change. Conversely, flow cytometry detects protein presence, including neuronal debris attached to glial cells. TUBB3, being a highly abundant neuronal cytoskeletal protein, can remain detectable even in fragmented material. As such, the protein signal measured by FACS is amplified by the presence of these fragments, leading to a higher apparent difference between CD56high and CD56⁻/CD56low populations. This highlights the inherent limitations in directly comparing mRNA and protein-level data, especially in heterogeneous populations like the ENS, where membrane remnants of one cell type are carried by other cell types, leading to a mismatch between cell identity and staining pattern. We have clarified these points in our previous paper [1], where this issue has been highlighted for the first time. In section 2.3, we have now revised the manuscript to briefly clarify this point and acknowledge the technical and biological reasons underlying this observed discrepancy (lines 348-361).
Comment Reviewer 2: Minor suggestion: the bibliography about the ENS is very old and could be updated with more recent papers (citations 1,3, and 8).
Authors’ response: We thank the reviewer for this valuable suggestion. In response, we have updated the manuscript's references to include more recent and relevant studies on the enteric nervous system.
References
- Windster, J.D.; Sacchetti, A.; Schaaf, G.J.; Bindels, E.M.; Hofstra, R.M.; Wijnen, R.M.; Sloots, C.E.; Alves, M.M. A combinatorial panel for flow cytometry-based isolation of enteric nervous system cells from human intestine. EMBO Rep 2023, 24, e55789.
- Drokhlyansky, E.; Smillie, C.S.; Van Wittenberghe, N.; Ericsson, M.; Griffin, G.K.; Eraslan, G.; Dionne, D.; Cuoco, M.S.; Goder-Reiser, M.N.; Sharova, T.; et al. The Human and Mouse Enteric Nervous System at Single-Cell Resolution. Cell 2020, 182, 1606-1622 e1623.